# CDL-UNet:Curriculum-Driven Lightweight 3D U-Net for Abdominal Organ Segmentation

Zhehao Wang[1,*], Xian Lin[1,*], Junjie Shi[1], Caozhi Shang[1], Li Yu[1], and Zengqiang Yan[1,†]

School of Electronic Information and Communications, Huazhong University of Science and Technology, Wuhan, China
`z_yan@hust.edu.cn`

**Abstract.** Deep learning has significantly advanced medical image segmentation, particularly for large-scale datasets. However, deploying these models on resource-limited devices like laptops presents two new difficulties: (1) Data-level challenges due to the large volume of data, limited availability of high-quality labels, and varying quality of pseudo-labels; and (2) Computational constraints, as CPU-only inference requires achieving both speed and performance under limited resources. To tackle these issues, we propose a novel curriculum-driven lightweight 3D U-Net (CDL-UNet) approach, which integrates a curriculum learning strategy, a label-based difficulty discriminator, and an adaptive sliding window inference method. Our curriculum learning strategy progressively trains the model with increasingly complex samples to enhance learning efficiency and accuracy. The label-based difficulty discriminator refines pseudo-labels and categorizes samples by difficulty, optimizing the training process. Finally, the adaptive sliding window inference ensures fast and accurate segmentation even with CPU-only hardware. Our method achieved an average score of 88.28% and 93.80% for organ DSC and NSD on the online validation set, with an average inference time of 38 seconds, demonstrating its effectiveness for high-quality segmentation on resource-constrained devices.

**Keywords:** Medical image segmentation · Curriculum learning · Computational efficiency

## 1 Introduction

Deep learning has revolutionized medical image segmentation through advanced models and strategies. However, deploying these models on resource-limited devices like laptops introduces significant challenges. Large-scale medical imaging datasets demand extensive computational power, which directly conflicts with the limited processing capabilities of such devices. This creates a fundamental

---

[1] †Equal contribution

[2] †Corresponding author

tension between achieving high precision and maintaining computational efficiency. Therefore, developing models that are both accurate and optimized for resource-constrained environments is essential for practical deployment.

The MICCAI FLARE 2022 challenge highlighted the potential to balance segmentation accuracy and efficiency on standard consumer-grade GPU hardware. In contrast, the MICCAI FLARE 2024 challenge introduces two new difficulties: (1) Data-level challenges: the dataset is characterized by its large volume, limited availability of high-quality labels, and varying quality of pseudo-labels; and (2) Computational constraints: the challenge requires model inference on CPU-only hardware in laptops, demanding even greater efficiency and lightweight design from the models.

To address these data-level challenges, it's crucial to consider the inherent complexity and varying quality of the data, which traditional machine learning methods often overlook. These methods typically rely on random sampling during training, leading to inefficiencies with large, complex datasets like those in FLARE 2024. As an alternative, curriculum learning(CL) has been proposed. First introduced by Bengio et al. [1], CL mimics human learning by starting with simpler concepts and gradually progressing to more challenging ones. This strategy has shown significant benefits in various tasks, including weakly-supervised object localization [20], object detection [21].Regarding the computational constraints, while FLARE 2022 [14] demonstrated the effectiveness of U-Net-based architectures on GPU hardware, these models face performance bottlenecks when deployed on CPU-only environments, particularly in resource-limited devices like laptops.

In this work, we propose a novel approach named Curriculum-Driven Lightweight 3D U-Net (CDL-UNet) to address these issues. To overcome the data-level challenges, we employ a curriculum learning strategy that progressively introduces the model to samples of increasing complexity. We enhance pseudo-label quality with a label enhancement module and use a difficulty discriminator to categorize samples into easy, hard, and erroneous groups. This staged training approach starts with simpler samples and incorporates more challenging ones over time. To meet the computational constraints, we develop a lightweight 3D U-Net architecture optimized for CPU inference and implement an adaptive sliding window inference strategy to further improve efficiency while maintaining accuracy.

Our contributions are summarized as follows:

- **Novel curriculum learning strategy:** We propose a difficulty-aware curriculum learning approach that effectively leverages noisy pseudo-labels to improve segmentation performance.
- **Lightweight 3D U-Net architecture:** We design a highly efficient 3D U-Net architecture optimized for CPU inference, capable of handling large-scale medical images.
- **Adaptive sliding window inference:** We develop an adaptive sliding window inference strategy that enables fast and accurate segmentation on CPU devices.

## 2   Method

### 2.1   Preprocessing

In our approach, we applied several preprocessing strategies to ensure the quality and consistency of the data used for training:

- **Data Cleaning**: We identified that some pseudo-labels contained noise, with label values falling outside the expected range of 0 to 13. To address this, we standardized these noisy labels by setting them to 0, effectively removing any erroneous label values and ensuring uniformity across the dataset.
- **Image Resampling**: Given the anisotropic nature of the imaging data, we resampled the images to a uniform spacing of [4.0, 1.2, 1.2]. This resampling step was crucial for normalizing the resolution across different dimensions, enabling the model to process the data consistently.

### 2.2   Proposed method

Our proposed method integrates several innovative strategies to enhance the performance and efficiency of medical image segmentation tasks. The core components include Data-level Curriculum Learning, a Label-based Difficulty Discriminator, a CPU-friendly lightweight 3D U-Net architecture, and Adaptive Efficient Sliding Window Inference.

**2.2.1 Data-level Curriculum Learning** In the MICCAI FLARE 2024 challenge, we applied a Data-level Curriculum Learning strategy to optimize our model's training. This strategy ranks the dataset by sample difficulty to enhance learning efficiency and performance. We categorized samples into three types based on pseudo-label quality and designed corresponding training stages.

High-quality pseudo-labels: These labels typically originate from simple samples that the model can predict accurately. Since the model's prediction ability is inversely related to the difficulty of the samples, pseudo-labels generated from simple samples tend to be of higher quality. These high-quality labels form the foundation of early-stage training, ensuring that the model firmly grasps basic concepts.

Moderate-quality pseudo-labels: As the model continues to learn, more complex samples are introduced, resulting in pseudo-labels of slightly lower quality. These challenging samples are crucial for the model to gradually adapt and overcome, reflecting its ability to handle increasingly complex situations.

Low-quality pseudo-labels: In some cases, even simple samples may produce very low-quality pseudo-labels due to the model's limitations rather than issues with the data itself.

Specifically, in the MICCAI FLARE 2024 challenge, we employed a difficulty discriminator based on label comparison to enhance each image's labels and assign a difficulty score. This scoring mechanism further categorizes the

samples into three groups: simple samples, difficult samples, and erroneous samples. Simple samples are those for which the model can generate high-quality pseudo-labels; difficult samples have slightly lower prediction quality, and erroneous samples represent cases where the model produces extremely low-quality pseudo-labels.

As illustrated in Fig. 1, our training strategy followed a two-phase approach: "Easy-Then-Hard." In the first phase (Easy Course), the model focused on simple samples, building a robust foundation with high-quality pseudo-labels for supervision. In the subsequent phase (Hard Course), attention shifted to difficult samples, enhancing the model's robustness and generalization. By training on increasingly complex and anomalous samples, we aimed to improve overall performance and maintain accuracy across different cases.

By integrating the difficulty discriminator and the customized "Easy-Then-Hard" dual-stage learning strategy, our method aimed to optimize the learning path of the model, leading to better performance and shorter training times. This approach not only improved the model's performance on the training set but also ensured stronger generalization on unseen data.

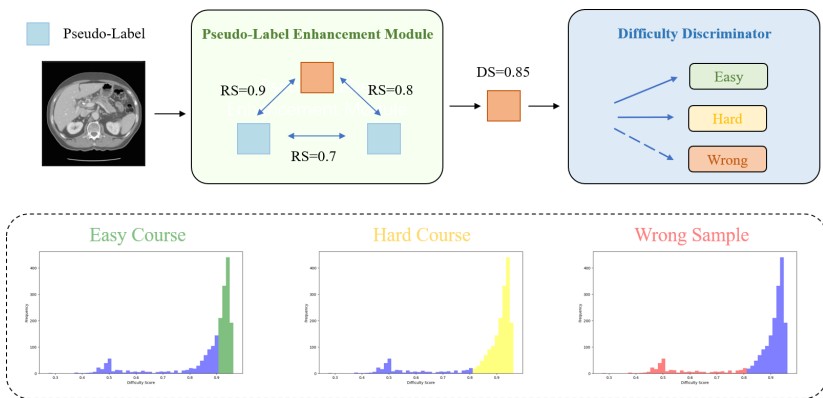

**Fig. 1.** Data-level Curriculum Learning.

**2.2.2 Label-based Difficulty Discriminator** The Label-Based Difficulty Discriminator module serves two primary functions: enhancing the pseudo-label quality for each image and calculating a difficulty score across different images.

**Pseudo-Label Enhancement Module** In the MICCAI FLARE 2024 challenge, two sets of pseudo-labels are provided for each image. However, it is challenging to determine which of the two labels is of higher quality. To address this, we introduce an additional reference label using the nnU-Net network as a third pseudo-label.

For any given image, the similarity between these pseudo-labels is considered a measure of mutual agreement. Specifically, if a pseudo-label has the highest

average similarity with the other two pseudo-labels, we consider it the most reliable. This can be formalized by computing the average Dice similarity, which we term the Reliability Score (RS), as follows:

$$\mathrm{RS}_i = \frac{\mathrm{Dice}(L_i, L_j) + \mathrm{Dice}(L_i, L_k)}{2}, \quad \text{for } i, j, k \in \{1, 2, 3\} \text{ and } i \neq j \neq k \quad (1)$$

Here, $\mathrm{RS}_i$ represents the Reliability Score for label $L_i$, indicating how closely it aligns with the other two labels. The label with the highest RS is selected as the most reliable pseudo-label.

**Difficulty Discriminator** After calculating the Reliability Scores using Equation 1, we can generate a Difficulty Score (DS) for each image to guide the curriculum learning process, as shown in Equation 2:

$$\mathrm{DS} = \frac{\mathrm{Dice}(L_{\mathrm{real}}, L_1) + \mathrm{Dice}(L_{\mathrm{real}}, L_2)}{2} \quad (2)$$

In this equation, $L_{\mathrm{real}}$ refers to the most reliable label selected from Equation 1. The Difficulty Score (DS) provides an estimate of the segmentation challenge for each image. A high DS implies that different models produce similar segmentation results, indicating an easier task. Conversely, a low DS suggests significant disagreement between the models, indicating a more difficult task. In extreme cases, where the DS is very low, it is likely that the segmentation contains significant errors, and such samples are excluded from training due to the absence of a reliable ground truth.

In summary, this method not only enhances the quality of pseudo-labels but also categorizes the data into different difficulty levels, facilitating a structured approach to curriculum learning.

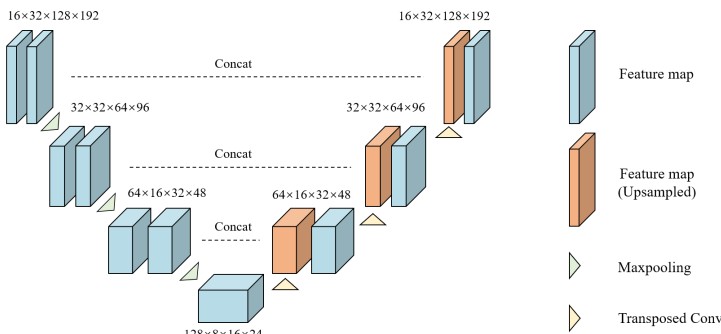

**Fig. 2.** Network architecture of Light 3D U-Net.

### 2.2.3 CPU-Friendly Lightweight 3D U-Net Architecture While 3D U-Net has demonstrated state-of-the-art performance in various medical image

segmentation tasks, deploying such a model for 3D medical image segmentation on a CPU poses significant challenges. To address this, we designed a CPU-friendly lightweight 3D U-Net architecture with the following key modifications, as illustrated in Fig. 2:

- **Smaller Patch Input:** We utilize a patch size of $32 \times 128 \times 192$, which is more suitable for CPU inference. Smaller patches reduce the computational load by limiting the number of voxels processed at once, allowing for more efficient memory management and faster processing times, crucial for CPU-based environments where memory and computational resources are limited.
- **Reduced Channel Count:** The initial number of channels is set to 16, which significantly lowers the model's complexity. By reducing the number of channels, we decrease the number of parameters and computational operations required during both training and inference, making the model more efficient without sacrificing segmentation accuracy.
- **Fewer Downsampling Layers:** Given the smaller patch size, we found that excessive downsampling was unnecessary. Therefore, we limited the model to three downsampling layers. This choice strikes a balance between maintaining spatial resolution and reducing computational complexity, ensuring that the model remains lightweight and effective for CPU-based segmentation tasks.

**2.2.4 Adaptive Efficient Sliding Window Inference** While the sliding window strategy provided by nnU-Net significantly improves prediction accuracy, it also introduces a substantial computational burden, particularly in CPU-limited environments. To address this issue, we propose an adaptive efficient sliding window inference method that leverages prior knowledge from Abdominal Organ CT images to greatly enhance inference efficiency.

- **Adaptive Window Sliding Strategy:** We observed that the dataset for MICCAI FLARE 2024 exhibits a considerable variation in image shape along the vertical axis (z-direction). For instance, some images have a very short z-axis length, such as Case_01300 (1, 24, 512, 512), while others have a significantly longer z-axis, such as Case_00947 (1, 997, 512, 512). Clearly, images with a shorter z-axis require fewer sliding window operations and impose a minimal computational load, allowing for a finer sliding window strategy to improve accuracy. Conversely, images with a longer z-axis require numerous sliding window operations, leading to a much higher computational load. Therefore, a coarser sliding window strategy can be adopted to enhance efficiency.
  The specific design is illustrated in Fig. 3. During inference, we first estimate the number of sliding windows using a fine-grained approach. If the number of sliding windows is below a threshold, we apply the fine-grained sliding window method. Otherwise, we switch to a coarser sliding window strategy to balance accuracy and computational efficiency.

– **Efficient Sliding Window:** Observations reveal that some Abdominal Organ CT images with a long z-axis only have a limited annotated region, with the majority being non-segmentation areas. Leveraging prior knowledge of human anatomy, the abdominal organs are expected to occupy a relatively small volume and are generally located in the central part of each transverse section.

Following the strategy proposed by [9], we first perform inference within the central window, as shown in Fig. 3 (highlighted in green). If this central window contains no foreground area, we can skip the surrounding windows, thus reducing unnecessary computations.

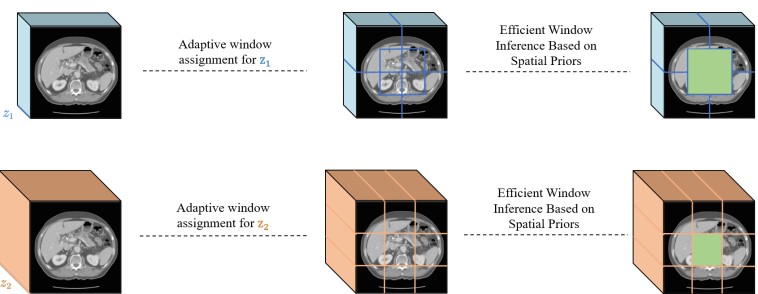

**Fig. 3.** Adaptive window sliding.

## 2.3 Post-processing

In our study, post-processing is employed to minimize false positives by eliminating small connected regions. This approach helps in refining the segmentation results by focusing on the most significant structures and reducing noise from minor isolated areas.

## 3 Experiments

### 3.1 Dataset and evaluation measures

The dataset is curated from more than 40 medical centers under the license permission, including TCIA [3], LiTS [2], MSD [19], KiTS [7,8], autoPET [6,5], AMOS [11], AbdomenCT-1K [18], TotalSegmentator [22], and past FLARE challenges [15,16,17]. The training set includes 2050 abdomen CT scans where 50 CT scans with complete labels and 2000 CT scans without labels. The validation and testing sets include 250 and 300 CT scans, respectively. The annotation process used ITK-SNAP [24], nnU-Net [10], MedSAM [12], and Slicer Plugins [4,13].

The evaluation metrics encompass two accuracy measures—Dice Similarity Coefficient (DSC) and Normalized Surface Dice (NSD)—alongside one efficiency

measures—runtime. These metrics collectively contribute to the ranking computation. During inference, GPU is not available where the algorithm can only rely on CPU.

## 3.2   Implementation details

**Environment settings**  The development environments and requirements are presented in Table 1.

**Table 1.** Development environments and requirements.

| System | Ubuntu 20.04.4 LTS |
|---|---|
| CPU | Intel(R) Xeon(R) Gold 6143 CPU @ 2.80GHz |
| RAM | $8 \times 32$GB |
| Programming language | Python 3.9 |
| Deep learning framework | Torch 2.3.1 |
| Specific dependencies | NVIDIA GeForce RTX 3090 (24G, $\times 1$) |
| Code | https://github.com/iam-nacl/FLARE2024 |

**Training Protocols**  In our approach, we adopted the default training protocols provided by the nnU-Net framework, which have been shown to be highly effective in medical image segmentation tasks.

- **Data Augmentation:** Extensive data augmentation was applied during the training process to improve the model's generalization capabilities. This includes operations such as elastic deformations, and intensity augmentations.
- **Patch Sampling Strategy:** We utilizes a sophisticated patch sampling strategy that balances between foreground and background regions. During training, patches are extracted from the images with a higher probability of containing foreground regions, ensuring that the model learns to accurately segment the structures of interest.
- **Optimal Model Selection Criteria:** The model used for inference is selected based on the final checkpoint after all training epochs

Further details are provided in Table 2

## 4   Results and discussion

### 4.1   Quantitative results on validation set

The results presented in Table  3 showcase the performance of our segmentation model across various abdominal organs. The high Dice Similarity Coefficient (DSC) and Normalized Surface Dice (NSD) values indicate the model's effectiveness in accurately segmenting most organs. Notably, the liver and spleen

**Table 2.** Training protocols.

| | |
|---|---|
| Network initialization | |
| Batch size | 2 |
| Patch size | 32×128×192 |
| Total epochs | 1000(Course easy)+1500(Course hard) |
| Optimizer | SGD |
| Initial learning rate (lr) | 0.01 |
| Lr decay schedule | 0.9 |
| Training time | 5h(Course easy)+8h(Course hard) |
| Loss function | Dice loss and cross entropy loss |
| Number of model parameters | 1.39M[1] |
| Number of flops | 64.02G[2] |
| $CO_2$eq | 2.19Kg[3] |

exhibited DSCs of 97.10% and 96.39%, respectively, demonstrating the model's proficiency in handling large, well-defined organs.

However, the performance on smaller and more complex structures, such as the right adrenal gland and gallbladder, was relatively lower, with DSCs of 82.25% and 80.75%, respectively. This discrepancy suggests that while the model performs exceptionally well on larger organs, there is room for improvement in segmenting smaller, more intricate structures.

On the online validation set, the model maintained robust performance, with an average DSC of 88.28%, highlighting its generalization capability across different datasets.

**Table 3.** Quantitative evaluation results on validation set.

| Target | Public Validation | | Online Validation | |
|---|---|---|---|---|
| | DSC(%) | NSD(%) | DSC(%) | NSD(%) |
| Liver | 97.10 ± 1.36 | 91.91 ± 5.42 | 96.92 | 97.32 |
| Right Kidney | 94.51 ± 13.70 | 93.84 ± 14.63 | 92.19 | 93.84 |
| Spleen | 96.39 ± 1.80 | 95.80 ± 5.18 | 94.06 | 95.16 |
| Pancreas | 89.99 ± 2.37 | 90.01 ± 5.12 | 85.95 | 96.08 |
| Aorta | 95.30 ± 1.90 | 98.49 ± 3.32 | 95.54 | 98.51 |
| Inferior vena cava | 91.91 ± 2.94 | 93.07 ± 4.61 | 90.75 | 92.93 |
| Right adrenal gland | 82.25 ± 7.54 | 95.93 ± 5.54 | 83.41 | 95.92 |
| Left adrenal gland | 83.17 ± 11.19 | 95.63 ± 9.64 | 82.36 | 94.49 |
| Gallbladder | 80.75 ± 30.51 | 83.14 ± 31.45 | 79.25 | 80.38 |
| Esophagus | 85.17 ± 5.61 | 90.71 ± 7.61 | 81.34 | 92.22 |
| Stomach | 93.34 ± 3.19 | 88.62 ± 7.41 | 92.13 | 94.76 |
| Duodenum | 86.92 ± 5.14 | 87.55 ± 6.09 | 80.12 | 92.88 |
| Left kidney | 93.34 ± 14.75 | 92.69 ± 16.08 | 93.61 | 94.95 |
| Average | 90.01 ± 7.19 | 92.11 ± 9.39 | 88.28 | 93.80 |

**Table 4.** Quantitative evaluation results on test set.

| Metric | Asian | | European | | North American | |
|---|---|---|---|---|---|---|
| | Mean | Median | Mean | Median | Mean | Median |
| DSC (%) | $86.2 \pm 6.9$ | 89.1 | $87.0 \pm 8.4$ | 89.6 | $87.4 \pm 4.4$ | 88.9 |
| NSD (%) | $92.0 \pm 6.2$ | 94.6 | $91.8 \pm 8.6$ | 95.3 | $92.5 \pm 4.9$ | 94.1 |
| Time (s) | $31.4 \pm 5.3$ | 30.6 | $30.6 \pm 8.1$ | 29.5 | $30.1 \pm 7.8$ | 29.3 |

Table 5 presents the results of the ablation study comparing the "Only Easy Course" training strategy with our final model, which integrates data-level curriculum learning. The final model consistently outperforms the "Only Easy Course" across most organs, particularly in challenging structures such as the right adrenal gland, where DSC improved from 78.64% to 82.25%, and NSD from 92.29% to 95.93%.

This demonstrates the effectiveness of our curriculum learning approach in enhancing model performance, especially for difficult-to-segment organs. The average DSC improvement from 88.67% to 90.01% and NSD from 90.34% to 92.11% further underscores the advantages of our comprehensive training strategy.

**Table 5.** Ablation studies on Public Validation: Performance Comparison of Only Easy Course vs. Final Model

| Target | Only Easy Course | | Final Results | |
|---|---|---|---|---|
| | DSC(%) | NSD(%) | DSC(%) | NSD(%) |
| Liver | 97.12 | 91.80 | 97.10 | 91.91 |
| Right Kidney | 93.65 | 92.49 | 94.51 | 93.84 |
| Spleen | 96.22 | 95.24 | 96.39 | 95.80 |
| Pancreas | 88.64 | 87.79 | 89.99 | 90.01 |
| Aorta | 95.08 | 98.17 | 95.30 | 98.49 |
| Inferior vena cava | 90.42 | 91.08 | 91.91 | 93.07 |
| Right adrenal gland | 78.64 | 92.29 | 82.25 | 95.93 |
| Left adrenal gland | 82.52 | 94.87 | 83.17 | 95.63 |
| Gallbladder | 76.17 | 78.16 | 80.75 | 83.14 |
| Esophagus | 83.39 | 88.39 | 85.17 | 90.71 |
| Stomach | 92.22 | 87.22 | 93.34 | 88.62 |
| Duodenum | 84.34 | 84.58 | 86.92 | 87.55 |
| Left kidney | 93.03 | 92.26 | 93.34 | 92.69 |
| Average | 88.67 | 90.34 | 90.01 | 92.11 |

### 4.2   Qualitative results on validation set

The qualitative results in Table 4 highlight both successful and challenging segmentation cases. In cases like FLARETs_0037 (slice 72) and FLARETs_0041 (slice 114), the final model successfully corrected misclassifications and improved

boundary precision compared to the initial "Easy Course" model. For instance, the final model accurately segmented the Right adrenal gland and Duodenum in Case FLARETs_0037, which the "Easy Course" model initially struggled with. However, in more challenging cases such as FLARETs_0048 (slice 76) and FLARETs_0032 (slice 172), both models faced difficulties, particularly with small or partially visible organs like the Right Kidney. Despite these challenges, the final model consistently demonstrated better overall segmentation, particularly in refining organ boundaries and reducing errors, underscoring the effectiveness of the curriculum learning approach.

### 4.3  Segmentation efficiency results on validation set

In Table 6, we report the efficiency evaluation results for segmentation on the validation set. The table presents the running time for various image sizes, illustrating the time required to process images of different dimensions. The results show that running time varies with image size. This evaluation highlights the impact of image complexity on processing time.

**Table 6.** Quantitative evaluation of segmentation efficiency in terms of the running time.

| Case ID | Image Size | Running Time (s) |
|---|---|---|
| 0059 | (512, 512, 55) | 36.25 |
| 0005 | (512, 512, 124) | 39.11 |
| 0159 | (512, 512, 152) | 33.27 |
| 0176 | (512, 512, 218) | 31.92 |
| 0112 | (512, 512, 299) | 45.49 |
| 0135 | (512, 512, 316) | 46.21 |
| 0150 | (512, 512, 457) | 48.67 |
| 0134 | (512, 512, 597) | 54.97 |

### 4.4  Results on final testing set

The qualitative results in Table 4 The model demonstrated consistent performance across regions (Asia/Europe/North America) with high segmentation accuracy (DSC: 86.2-87.4%, NSD: 91.8-92.5%) and efficient inference times (29.3-31.4s), showing minimal regional variability.

### 4.5  Limitation and future work

While our approach performed well overall, it faced challenges in segmenting smaller or less distinct structures like the adrenal glands and gallbladder. Additionally, cases with atypical anatomy or low-contrast boundaries led to minor

inaccuracies. Future work will aim to improve the model's handling of these challenges by incorporating advanced data augmentation techniques and integrating multi-scale features. Refining our curriculum learning strategy could also enhance segmentation accuracy in these complex scenarios.

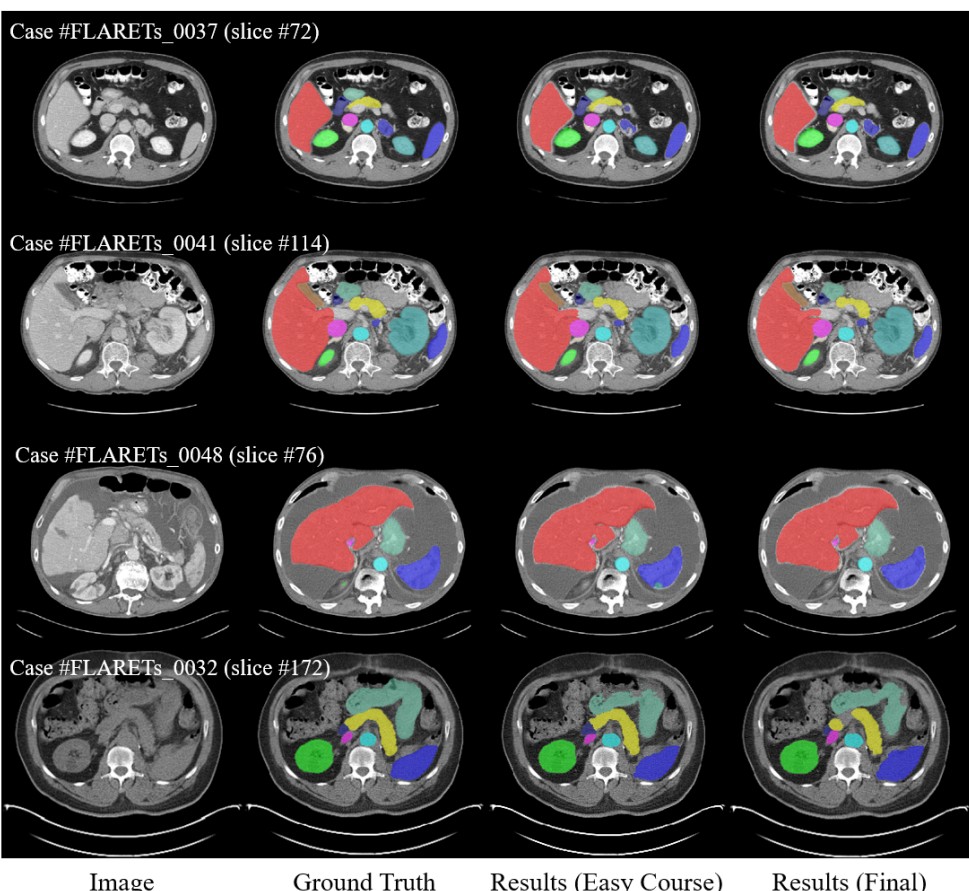

**Fig. 4.** Qualitative results on validation set.

## 5   Conclusion

In this study, we proposed a comprehensive approach to abdominal organ segmentation, leveraging a data-level curriculum learning strategy alongside a CPU-friendly lightweight 3D U-Net architecture. Our method effectively addressed the challenges of segmenting both large and small structures by prioritizing easier

cases in early training phases, thereby enhancing model generalization. Quantitative results demonstrated that our approach achieved high accuracy across multiple organs, particularly excelling in the segmentation of major organs such as the liver and spleen. Despite some limitations in segmenting smaller or less distinct organs, our approach shows promise for future refinement and application in clinical settings.

**Acknowledgements** The authors of this paper declare that the segmentation method they implemented for participation in the FLARE 2024 challenge has not used any pre-trained models nor additional datasets other than those provided by the organizers. The proposed solution is fully automatic without any manual intervention. We thank all data owners for making the CT scans publicly available and CodaBench [23] for hosting the challenge platform.

## Disclosure of Interests

The authors declare no competing interests.

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

**Table 7.** Checklist Table. Please fill out this checklist table in the answer column.

| Requirements | Answer |
| --- | --- |
| A meaningful title | Yes |
| The number of authors (≤6) | 6 |
| Author affiliations and ORCID | Yes |
| Corresponding author email is presented | Yes |
| Validation scores are presented in the abstract | Yes |
| Introduction includes at least three parts: background, related work, and motivation | Yes |
| A pipeline/network figure is provided | Figure 1 |
| Pre-processing | Page 3 |
| Strategies to improve model inference | Page 5,6,7 |
| Post-processing | Page 7 |
| The dataset and evaluation metric section are presented | Page 7 |
| Environment setting table is provided | Table 1 |
| Training protocol table is provided | Table 2 |
| Ablation study | Page 10 |
| Efficiency evaluation results are provided | Table 5 |
| Visualized segmentation example is provided | Figure 4 |
| Limitation and future work are presented | Yes |
| Reference format is consistent. | Yes |