# OpenReview forum: "CDL-UNet:Curriculum-Driven Lightweight 3D U-Net for Abdominal Organ Segmentation"
_MICCAI.org/2024/Challenge/FLARE — FLARE 2024 withMinorRevisions_

### Official Review · Reviewer_CMV9 · 2025-01-20
**A CPU-Efficient Curriculum Learning Approach for Abdominal Organ Segmentation**

**Rating:** 8
**Confidence:** 4

**Review:**

This paper presents CDL-UNet, a novel CPU-only abdominal organ segmentation approach for the FLARE 2024 challenge. The method integrates curriculum learning for progressive training, a label-based difficulty discriminator, a lightweight 3D U-Net architecture, and an adaptive sliding window inference strategy. It achieves 88.28% DSC and 93.80% NSD on the validation set with a 38-second inference time, demonstrating effective segmentation performance under CPU-only constraints.

My only concern is that the image size in the fourth row of Figure 4 should be consistent with the first three rows.

---

> ### Author Response · Authors · 2025-03-29
> **Official Comment by Authors**
>
> Thank you for your comment.. In the revised manuscript, we have resized case0032 in Figure 4 to ensure better visual consistency across all cases

---

### Official Review · Reviewer_6bke · 2025-01-23
**Good Paper**

**Rating:** 9
**Confidence:** 4

**Review:**

This paper proposes a curriculum learning-driven lightweight 3D U-Net, named CDL-UNet, to address the challenges of abdominal organ segmentation under data and computational resource constraints. The innovation of the paper lies in the introduction of a data-level curriculum learning strategy, which uses a difficulty discriminator to classify training data and gradually transitions from simple to complex samples, thereby improving the model's learning efficiency and generalization capability. Moreover, the paper designs an optimized lightweight 3D U-Net architecture specifically for CPU environments and introduces an adaptive sliding window inference strategy that significantly reduces inference time while maintaining high segmentation accuracy.
However, the fourth-row image in Figure 4 is too small.

---

> ### Author Response · Authors · 2025-03-29
> **Official Comment by Authors**
>
> Thank you for your suggestion. In the revised manuscript, we have resized case0032 in Figure 4 to ensure better visual consistency across all cases

---

### Official Review · Reviewer_KYTr · 2025-01-27
**A Very Complete Paper**

**Rating:** 8
**Confidence:** 4

**Review:**

This article introduces a method named CDL-UNet for abdominal organ segmentation. The method aims to address the challenges of deploying deep learning models on resource-limited devices such as laptops. CDL-UNet combines a curriculum learning strategy, a label-based difficulty discriminator, and an adaptive sliding window inference method to improve segmentation efficiency and accuracy. The method achieved an average organ DSC of 88.28% and NSD of 93.80% on the online validation set, with an average inference time of 38 seconds, demonstrating its effectiveness for high-quality segmentation on resource-constrained devices. However, the fourth-row image in Figure 4 is too small.

---

> ### Author Response · Authors · 2025-03-29
> **Official Comment by Authors**
>
> Thanks for your suggestion. In the revised manuscript, we have resized case0032 in Figure 4 to ensure better visual consistency across all cases

---

### Official Review · Reviewer_hErM · 2025-03-11
**Typos and style**

**Rating:** 9
**Confidence:** 5

**Review:**

Please resize case0032 in Figure 4;

---

> ### Author Response · Authors · 2025-03-29
> **Official Comment by Authors**
>
> We appreciate the reviewer's suggestion. In the revised manuscript, we have resized case0032 in Figure 4 to ensure better visual consistency across all cases

---

### Decision · Program_Chairs · 2025-03-20

**Decision:**

Accept

**Comment:**

Please carefully address the reviewers' comments in the revision.

---

> ### Author Response · Authors · 2025-03-29
> **Official Comment by Authors**
>
> Thank you for your comment.We have carefully addressed the reviewers' comments in the revision.